# "Cool" Roofs as a Heat-Mitigation Measure in Urban Heat Islands: A Comparative Analysis Using Sentinel 2 and Landsat Data

**Terence Mushore [1,2], John Odindi [1,]\* and Onisimo Mutanga [1]**

[1]  Discipline of Geography, School of Agricultural, Earth and Environmental Sciences, University of KwaZulu-Natal, Scottsville, Pietermaritzburg 3209, South Africa
[2]  Department of Space Science and Applied Physics, Faculty of Science, University of Zimbabwe, 630 Churchill Avenue, Mt Pleasant, Harare 00263, Zimbabwe
\*  Correspondence: odindi@ukzn.ac.za

**Abstract:** Urban growth, characterized by expansion of impervious at the cost of the natural landscape, causes warming and heat-related distress. Specifically, an increase in the number of buildings within an urban landscape causes intensification of heat islands, necessitating promotion of cool roofs to mitigate Urban Heat Islands (UHI) and associated impacts. In this study, we used the freely available Sentinel 2 and Landsat 8 data to determine the study area's Land Use Land Covers (LULCs), roof colours and Land Surface Temperature (LST) at a 10-m spatial resolution. Support Vector Machines (SVM) classification algorithm was adopted to derive the study area's roof colours and proximal LULCs, and the Transformed Divergence Separability Index (TDSI) based on Jeffries Mathussitta distance analysis was used to determine the variability in LULCs and roof colours. To effectively relate the Landsat 8 thermal characteristics to the LULCs and roof colours, the Gram–Schmidt technique was used to pan-sharpen the 30-m Landsat 8 image data to 10 m. Results show that Sentinel 2 mapped LULCs with over 75% accuracy. Pan-sharpening the 30-m-resolution thermal data to 10 m improved the spatial resolution and quality of the Land Surface map and the correlation between LST and Normalized Difference Vegetation Index (NDVI) used as proxy for LULC. Green-colour roofs were the warmest, followed by red roofs, while blue roofs were the coolest. Generally, black roofs in the study area were cool. The study recommends the need to incorporate other roofing properties, such as shape, and further split the colours into different shades. Furthermore, the study recommends the use of very high spatial resolution data to determine roof colour and their respective properties; these include data derived from sensors mounted on aerial platforms such as drones and aircraft. The study concludes that with appropriate analytical techniques, freely available image data can be integrated to determine the implication of roof colouring on urban thermal characteristics, useful for mitigating the effects of Urban Heat Islands and climate change.

**Keywords:** cool roofs; urban heat islands; land surface temperatures; roof colour; mitigation; urban growth

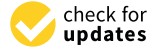



## 1. Introduction

Urbanization, and the associated urban land use and land cover (LULC) spatial structure transformations influence the urban thermal characteristics [1–3]. This process is typified by transformation from natural to impervious surfaces such as buildings and other urban fabrics that alter surface and near-surface temperatures [4,5]. The increase in temperatures attributable to urban growth are associated with a range of challenges that include adverse effects on human health, increased water and energy demand and air pollution [6–8]. As such, urbanization and consequent thermal elevation has been known to exacerbate in- and out-door ambient thermal discomfort that diminish the quality of urban

life [9,10]. Hence, it is increasingly becoming desirable to adopt climate-smart approaches that could enhance sustainable urban living.

Remotely sensed data offer an opportunity to determine urban spatio-temporal variations and their respective thermal characteristics [11,12]. Additionally, remotely sensed data allows for analysis at a range of time-scales that include sub-seasonal patterns. Over the years, technological advancement has facilitated the acquisition of both optical and thermal data on the same sensor platforms (e.g., Landsat, ASTER and MODIS), valuable for urban landscape transformation and thermal analysis [13–16]. Hence, these moderate resolution sensors have been widely used to determine the influence of LULCs on the thermal environment e.g., [17–22]. However, such moderate resolution datasets suffer from the mixed pixel problem, especially in urban areas characterized by landscape heterogeneity, which compromises their value for detailed surface analysis such as the detection of individual houses and their thermal properties [23].

Fortunately, recent sensor developments and advancements in computational power offer an opportunity for improved land surface analysis. For instance, whereas the Landsat series has over the years improved in spectral and radiometric properties, new generation sensors such as Sentinel 2 offer data with improved spatial resolution [24–26]. The sensors' 10-m spatial resolution for instance, ref. [27] allows for analysis of complex environments such as urban areas with reduced mixed pixel effect and high mapping accuracy. Whereas the Sentinel 2s platform lacks a thermal sensor, its integration with high quality data such as Landsat has potential to improve our knowledge of the relationship between urban LULCs and surface temperatures. Recently, Mushore et al. [28] showed that pan-sharpening of Landsat thermal data improves its Land Surface Temperature (LST) mapping accuracy, while Kaplan and Avdan [26] used Sentinel 2's pan-sharpened 10-m to improve 20-m resolution bands. However, whereas the Sentinel's 10-m spatial resolution optical data can be used to derive detailed urban surface features, Landsat thermal data need to be at a similar spatial resolution for optimal analysis and mapping accuracy.

Several studies have demonstrated that built-up areas absorb and store large amounts of heat when compared to other LULC types, e.g., [22,27–30]. The thermal effect is enhanced by increased building densities that result in large surface areas for heat absorption. Furthermore, dense high-rise buildings increase heat storage capacity as walls present even larger surface areas for heat absorption. Buildings also concentrate heat in an area by retarding its removal by winds [31]. To date, a significant number of studies have dwelt on the effect of buildings on temperature. For instance, the effect of building density and height have been widely demonstrated in both the developed and the developing world, e.g., [28,32–35]. Besides density and height, building materials and other properties such as roof characteristics influence a built environment's thermal properties. For example, Mackey et al. [36] demonstrated that cool roofs surpassed green roofs, street trees and green spaces in cooling effects in Chicago. However, the adoption of remotely sensed data to understand the influence of roofing properties on temperature remains limited. Emphasis has been largely placed on understanding the influence at a broad scale and general LULC classes on the thermal environment. Focus on localized phenomena that include the effect of individual houses and their characteristics such as roof properties using freely available remotely sensed data has remained a grey area.

Studies on the effect of roofs on buildings thermal characteristics have mainly focused on rooftops with vegetation (i.e., 'green roofs') and commonly use data derived from installed meteorological instruments and analytical models [37–40]. Other studies have investigated roof characteristics such as roof angle; for instance, Tian et al. [41] compared the thermal characteristics of curved and flat roofs. Studies on roof colour have established that white roofs have more cooling effect than grey, red and black roofs [42–44], while coating coloured roofs with highly reflective materials can increase thermal performance and energy efficiency of buildings [45]. For instance, Libbra et al. [45] found that the use of cool roofs can reduce air conditioning energy consumption by 70%. For the same roof type, variations such as colour and age may also influence their interactions with heat [43,46,47].

However, due to limitations of the new generation sensors' spatial resolution, literature on the influence of roof colour on thermal performance of buildings remains scarce.

Zhao et al. [48] examined daytime and nighttime effects of roof footprints and configurations using high resolution airborne LIDAR and Quickbird satellite data (2.4-m resolution) and MODIS/ASTER simulated airborne 7-m-resolution surface temperature data. They observed that rooftop spectral attributes, slope, aspect and surrounding trees affected roof surface temperature. Although they accurately delineated roof configurations, they did not segment the roofs by colour. Furthermore, while sensors on aerial platforms such as drones and airplanes can provide data for detailed analysis of effects of roofs on thermal characteristics, such data remain expensive and not viable for studies over large spatial extents. Hence, there is a need to test the value of freely available moderate resolution optical and thermal datasets to enhance our understanding on the influence of building roof colour on thermal characteristics, especially in growing cities of developing countries. Such efforts are necessary to determine the potential adoption of roof type and colour to mitigate heat islands.

According to Alchapar and Correa [46], roof coating is the most influential morphological determinant of roof thermal behavior, while Libbra et al. [45] notes that roof colour controls the absorption of heat during the day and its emission at night. As such, it is necessary to consider "cool" roofs for UHI mitigation. Hence, in relation to adjacent LUCLs, this study sought to determine the value of Sentinel 2 10-m resolution and pan-sharpened Landsat image data in differentiating the influence of roof colour on surface thermal values.

## 2. Methodology

### 2.1. Description of the Study Area

The study was carried out in a low-density residential area close to the Central Business District (CBD) of the capital city of Zimbabwe, Harare (Figure 1). Since the study sought to determine the influence of roof colour on urban thermal characteristics, it was restricted to a small spatial extent to limit excessive heterogeneity that typifies urban landscapes. Also, a large area could have introduced additional variables (e.g., elevation and slope) that influence thermal characteristics. The area is in a low-to-medium-density residential type, however, some of the houses, especially towards the CBD, have been turned into offices. Low-to-medium-density residential areas in Zimbabwe are characterized by spacious housing units, high land value and higher vegetation density when compared to high-density residential areas, which are predominantly occupied by the low-income strata. Since house units in the low-to-medium-density residential areas are generally large, they are potentially discriminable using 10-m or higher spatial resolution image data. Hence, based on the 10-m spatial resolution image data, the area was chosen to minimize the mixed pixel problem that characterizes the high-density residential areas. Furthermore, the area is dominated by houses with tiles, thus eliminating the effect of other roof types such as concrete, zinc, aluminium or thatch on the area's thermal characteristics. This enabled the study to determine the variability in temperature based on roof tiling of different colours.

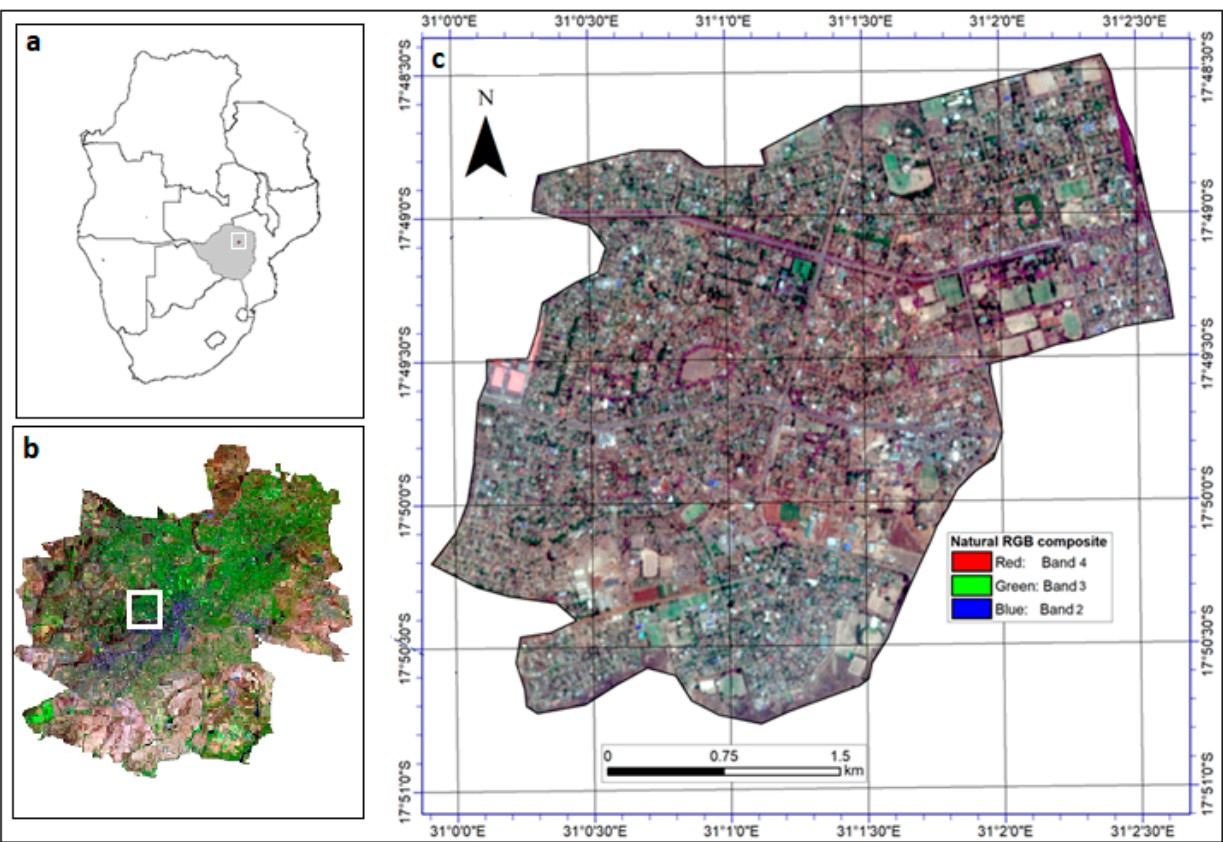

**Figure 1.** Location of the study area in Southern Africa, Zimbabwe and Harare (**a**), Harare and the study area (**b**) and 10-m resolution natural colour composite showing variations of LULC in the study area (**c**).

### 2.2. Field and Remotely Sensed Datasets

A field survey was conducted to identify the LULCs and roof colours in the area. The survey revealed that the major LULC classes are grasslands, buildings with heterogeneous tiled roof colouring, bare soil and roads. The building class was further split into roof colours in line with the main objective of the study, and a stratified random sampling approach used to collect the ground control points (GCPs). Non-tiled roofs were categorized into "Other LULCs". For each identified category, coordinates of representative covers were collected using a handheld Global Positioning System. To maximize the spectral and thermal variability, the hot dry season (mid-September to mid-November) was chosen for the collection of the well-distributed LULCs' GCPs as it presents a period of maximum solar energy with no rainfall cooling effect. The LULC types were verified using a GoogleEarth image, which was also used to verify the roof colours and to shift the GCPs to the roof center for classification and validation purposes. The data were split: 70% to be used for classification and 30% for validation.

Landsat and Sentinel 2 data were downloaded from the United States Geological Survey's earth explorer portal at no cost. To minimize variation between field data and image scenes, cloud-free imagery was collected on dates close to field data collection. Two Landsat images (scene capture dates: 16 September 2021 and 3 November 2021) and Sentinel images (scene capture dates: 18 September and 2 November 2021) were used in this study. The dates were chosen as they correspond to the period of maximum radiation in the hot season and the proximity of the two sensor dates. The wind was calm and cloudless, presenting similar weather conditions when data from the same sensor were acquired. Given that the period is dry, vegetation conditions were assumed to be uniform and largely maintained by irrigation/watering throughout the periods. Landsat data acquired on the 16th of September were matched with Sentinel data of the 18th of September, a short enough

period to assume that LULCs did not change. Similarly, Landsat data acquired on the 3rd of November were matched with Sentinel data of the 2nd of November. This gave the best compromise to enable relating and blending multi-sensor data with different spatial and temporal resolutions. The two Landsat images were used to compute the average LST to minimize randomness associated with a single-date image, while the two Sentinel 2 image datasets were used for LULC classification. Multi-spectral optical 10-m resolution Sentinel 2 and 3-m resolution Landsat 8 data for each acquisition date were merged into a multi-layer files using the 'Layer stacking' tool in ENVI software. This was done separately for Landsat 8 and Sentinel 2 data. In order to eliminate the effect of aerosols on reflectance values, atmospheric correction was done using the Fast Line-of-sight Atmospheric Analysis of Spectral Hypercubes (FLAASH) module in the ENVI software. Due to the proximity to the CBD and location in an urban area, the urban aerosol mode was used in FLAASH, which produced multi-layer reflectance files. Multilayer 10-m resolution Sentinel 2 reflectance data were required to provide multi-spectral information to enhance separation of features in supervised image classification. On the other hand, spectral reflectivity bands in the near-infrared and red range from Landsat 8 were needed for the computation of normalized difference vegetation index (NDVI), useful for emissivity correction in LST retrieval.

### 2.3. Separability Analysis

Sentinel 2 10-m resolution bands and the 70% of the field-collected GPS points for the LULCs and roof colour categories were overlaid in an ENVI version 4.7 environment. Surface separability was done using the Transformed Divergence Separability Index (TDSI) based on Jeffries Mathussitta distance analysis. For each paired classes, TDSI ranges between 0 and 2, with values greater than 1 indicating that two classes are distinguishable and values close to 2 implying very high separability. Values below 1 and close to 0 suggests that the classes should be merged. The TDS analysis was necessary to test whether different roof colours and LULCs could be separated before classification.

### 2.4. Land Use/Cover Mapping and Retrievals of Roof Colours

The LULCs were derived from Sentinel 2's 10-m bands based on the 70% GCPs using the Support Vector Machines (SVM) algorithm in ENVI version 4.7 software. Default settings of 0.083 and 100 were used for Gamma in kennel function and penalty parameter, respectively. The SVM uses two classes of training samples within a multidimensional feature space to fit an optimum dividing hyperplane. It aims to maximize the variability between the most proximal training samples (support vectors) and the hyperplane [49,50]. To achieve our objective, we chose a Gaussian radial-basis kennel function as it is ideal for working in an infinite-dimensional space and has a single parameter [49–51]. We classified the images into eight classes, namely, Roads and Bare, Trees, Grassland, Red roof, Blue roofs, Green colour roofs, Black roofs and Grey roofs. To display the roof colours from other LULCS, the Roads and Bare, Trees and Grassland classes were amalgamated into "Other LULC". Thereafter, a confusion matrix was generated. A confusion matrix compares the assigned class labels on the classified map with the location's actual LULC class observed in the field (ground truth). The confusion matrix was used to derive the most widely used accuracy indicators, namely, Overall Accuracy (OA) and Kappa (k) [52].

### 2.5. Land Surface Temperature Retrieval from Landsat 8 Data

Band 10 of Landsat 8 was used to retrieve LST from thermal infrared data using Planck's radiation law-based equation for single-channel Landsat thermal data [53]. Initially, thermal infrared digital numbers were converted to surface-leaving radiance using Equation (1);

$$L_I = M_I\, Q_{CAL} + A_L \tag{1}$$

where, $L_l$ is spectral radiance at Top of the Atmosphere measured in Watts/m$^2$/srad/μm, $M_l$ is Band-specific multiplicative rescaling factor, $Q_{CAL}$ represents pixel values (Digital Numbers) and $A_L$ is the Band-specific additive rescaling factor. $M_l$, $A_L$ and $Q_{CAL}$ are

obtained from the metadata downloaded together with the Landsat 8 data. As described by U.S. Geological Survey [54], the coefficients for converting digital numbers to thermal radiances were obtained from the metadata file accompanying Landsat 8 data download.

Mumtaz et al. [55] provides an in-depth description of steps for land surface temperature retrieval. The procedures include conversion of thermal radiances to blackbody/brightness temperature followed by emissivity correction to obtain surface temperatures. As such, derived radiances were used in Equation (2) to determine brightness/blackbody temperature.

$$T_B = \frac{K_2}{ln\left(\frac{K_1}{L_{II}} + 1\right)} \tag{2}$$

where, $T_B$ is the brightness temperature (in degrees Kelvins), $K_2$ and $K_1$ area conversion constants for the thermal band (in this case Band 10), also obtained from the metadata file. Since brightness temperature over surfaces is calculated by assuming emissivity to be equal to 1, further analysis must consider actual emissivity which varies with LULC type. This was achieved through emissivity correction, which converted brightness temperatures to actual surface temperatures using Equation (3) [53,55].

$$T_S = \left(\frac{T_B}{1 + \left(\frac{\lambda \times T_B}{\alpha}\right) ln\ \varepsilon}\right) - 273.16 \tag{3}$$

where $T_S$ is the LST in Degree Celsius, $\lambda$ is the central wavelength of emitted radiance (10.9 μm for band 10 of Landsat 8), $\varepsilon$ is the emissivity and $\alpha$ is a constant ($1.438 \times 10^{-2}$ mK). Due to its simplicity, Equation (4) was used to estimate emissivity from Normalized Difference Vegetation Index (NDVI) using [55–57];

$$\varepsilon = a + b\ ln(NDVI) \tag{4}$$

where $a$ = 1.0094 and $b$ = 0.047. Developed in Botswana, which is close to the study area, the equation was chosen due to ease of computation, parsimony and proven applicability in a tropical environment [55]. The *NDVI* was retrieved using reflectance in the Near Infrared (Band 5) and Red (Band 4) of Landsat 8 in Equation (5) [53,58];

$$NDVI = \frac{(NIR - RED)}{(NIR + RED)} \tag{5}$$

where *NIR* and *RED* are reflectance in the near-infrared and red ranges [59] derived from Band 5 and Band 4 of Landsat 8, respectively. The steps above obtained LST at a resampled resolution of 30 m, requiring further enhancement for analysis of roofs thermal properties at a local scale.

### 2.6. Gram-Schmidt Pan-Sharpening Based Method for LST Image Data Pan-Sharpening

Improvement of LST data from 30-m to 10-m spatial resolution was achieved using the Gram–Schmidt pan-sharpening technique. The Gram–Schmidt method uses weighted addition of multi-spectral bands to produce a replicated pan-sharpened low-resolution image. Gram–Schmidt orthogonalization is then used to make all bands of the multi-spectral low-resolution data orthogonal and scalar products are computed and turned into covariances [60]. For each band of the low-resolution multispectral data, angles between the band and the simulated low-resolution panchromatic are computed. Gain and bias of the high-resolution panchromatic band is used to simulate each low-resolution panchromatic band. The process is reversed using the same transformation coefficients, and high resolution pan-sharpened bands are produced [60,61]. Using Gram–Schmidt transformation, the colours of the composite RGB pan-sharpened bands are near similar to the respective original images, thus there is minimal distortion of spatial patterns. The method was chosen because all transform coefficients are computed in the low MS

resolution, hence are more robust to spatial misalignment of the bands than most other pan-sharpening methods [60]. In this study, the Sentinel 2's 10-m resolution Band 2 was used to improve the Landsat data. The purpose was mainly aimed at producing thermal data for retrieval of LST at 10-m resolution to match with the products from supervised image classification.

### 2.7. Intensity Analysis for In-Depth Characterization of Local Climate Zones Changes

The LST spatial configurations before and after pan-sharpening were compared to assess the effect of improving spatial resolution on image quality. The root mean-square error was also used to check the difference after resampling the LSTs to 30-m resolution to assess the effect on values per pixel. A 30-m resolution Landsat scene was used to derive NDVI and its correlation with 30-m LST (after resampling using a bicubic convolution) was obtained using the "Zonal Statistics as a Table" tool in ArcGIS version 10.2, ESRI, Redlands, California, USA. Similarly, NDVI was calculated using 10-m resolution near infrared and red Sentinel 2 and correlated with pan-sharpened 10-m resolution LST. The LST correlations with NDVI before and after pan-sharpening were then compared.

### 2.8. Linking LULC Types and Roof Colours with LST

Qualitatively, the spatial structure of LULC and roof colours was compared with that of LST using visual inspections of maps produced from the combination of Sentinel 2 10-m resolution and Landsat 8 thermal data. For quantitative assessment, field-collected points corresponding to each LULC and roof colour category were used to extract LST values using the "Extract values to points" spatial overlay function in ArcGIS version 10.2. The field-collected points were used instead of overlaying the LST with the retrieved LULC map to eliminate the effect of classification accuracy on extracted temperatures for the different categories. Box plots were used to depict the variations of LST between and within LULC and roof colour categories in the study area. The mean LSTs for the different LULC and roof colour categories were also used to compare their thermal performances. This was done to assess the effect of improving resolution on the relationship between LULC and LST using NDVI as a proxy for LULC spatial patterns.

Figure 2 summarizes the procedures from data collection to linking of roof colours to LST spatial structures in the study area.

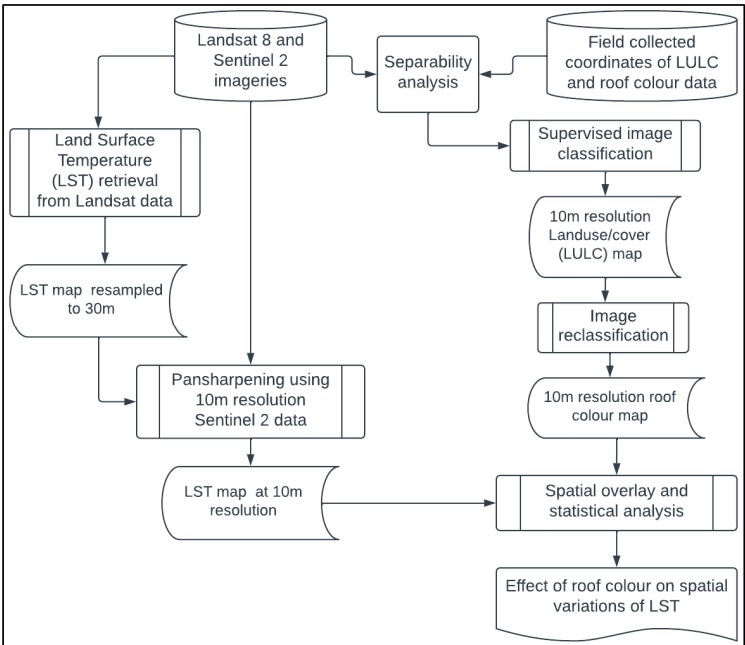

**Figure 2.** Summary of steps followed in the study.

## 3. Results

### 3.1. Separability of LULC and Roofs by Colour

Table 1 indicates that the TDSI values ranged between 1.74 and 2.00, implying that the LULC and roof colour categories were distinguishable using spectral signatures from 10-m resolution Sentinel 2 bands. Tarred roads and Trees were the most discriminable classes while the Green and Grey roofs were least discriminable, as indicated by TDSI values of 2.000 and 1.708, respectively. However, although Green and Grey roofs were the least separable, the TDSI value was significantly above the separability threshold of 1, hence guaranteed that the two classes were distinguishable. Among the roof colour categories, blue and red roofs were the most separable with a TDSI value of 1.995. The trees LULC category was the most separable from other cover types, with TDSI values ranging between 1.997 and 2.000. Overall, TDSI values greater than 1.7 indicate that the LULC and roof colour classes in the study area were easily distinguishable.

**Table 1.** Discriminability of LULC types in the study area using 10-m resolution Sentinel 2 data.

| Compared LULC and Roof Classes | TDSI |
|---|---|
| Green-colour roofs and Grey roofs | 1.708 |
| Black roofs and Grey roofs | 1.769 |
| Black roofs and Green-colour roofs | 1.826 |
| Grey roofs and Red roofs | 1.835 |
| Black roofs and Tarred roads | 1.874 |
| Black roofs and Red roofs | 1.920 |
| Green-colour roofs and Red roofs | 1.928 |
| Blue roofs and Grey roofs | 1.930 |
| Red roofs and Bare areas | 1.935 |
| Blue roofs and Green-colour roofs | 1.944 |
| Grey roofs and Tarred roads | 1.945 |
| Black roofs and Blue roofs | 1.955 |
| Grey roofs and Bare areas | 1.965 |
| Black roofs and Bare areas | 1.970 |
| Grass and Bare areas | 1.972 |
| Grass and Red roofs | 1.985 |
| Red roofs and Tarred roads | 1.985 |
| Grass and Grey roofs | 1.986 |
| Green-colour roofs and Bare areas | 1.990 |
| Blue roofs and Tarred roads | 1.994 |
| Blue roofs and Red roofs | 1.995 |
| Black roofs and Trees | 1.996 |
| Grass and Green-colour roofs | 1.997 |
| Black roofs and Grass | 1.998 |
| Blue roofs and Bare areas | 1.998 |
| Grass and Tarred roads | 1.998 |
| Grey roofs and Trees | 1.999 |
| Blue roofs and Grass | 1.999 |
| Tarred roads and Bare areas | 2.000 |
| Trees and Bare areas | 2.000 |
| Blue roofs and Trees | 2.000 |
| Green-colour roofs and Trees | 2.000 |
| Grasslands and Trees | 2.000 |
| Red roofs and Trees | 2.000 |
| Tarred roads and Trees | 2.000 |

### 3.2. Land Use/Cover and Roof Colour Mapping Using 10 M Resolution Sentinel 2 Data

The LULCs presented the houses surrounded by abundant vegetation, a characteristic of low-to-medium-density residential areas in Zimbabwe (Figure 3a). The study area has large grasslands, especially in the northeastern regions. The grasslands in the northeast are mainly sporting grounds. The other open grasslands within built-up areas are school

grounds while the fragmented grasslands are mainly lawns around houses as well as unused land. The other abundant vegetation was in built-up areas. Figure 3b shows that due to narrow widths in relation to the 10-m resolution of the data, most roads, especially in the black roofs category were not visible.

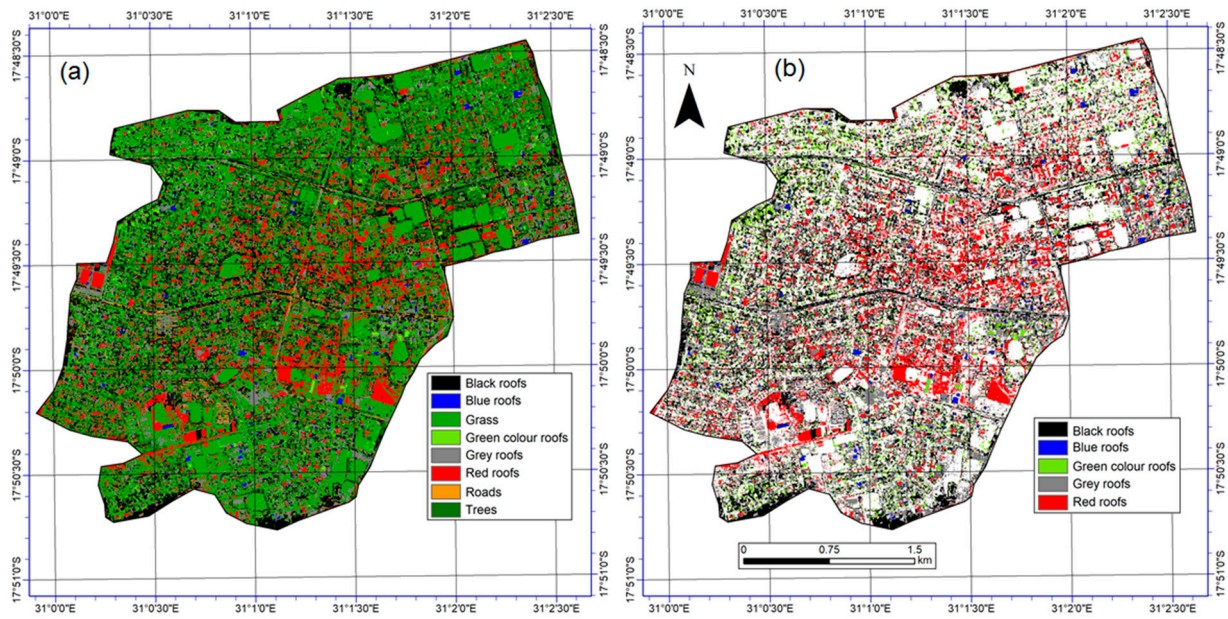

**Figure 3.** 10-m resolution (**a**) LULC map and (**b**) roof colour map.

### 3.3. Accuracy of LULC and Roof Colour Retrievals from Sentinel 2 Data

LULC and roof colour categories were mapped with Overall Accuracy (OA) of 84.5% and Kappa of 0.81. Producer Accuracies (PA) were greater than 75% except for the grey roofs and tarred roads (Table 2). User Accuracies (UA) were less than 75% for the black roofs, trees and grey roofs, while greater than 77% for the other categories. The red roofs were mapped with the highest accuracy of all the other categories (PA and OA greater than 93%).

**Table 2.** LULC and roof colour mapping accuracies.

| LULC and Roof Colour Category | Producer Accuracy (%) | User Accuracy (%) |
| --- | --- | --- |
| Black roofs | 74.44 | 70.42 |
| Blue roofs | 95.79 | 98.45 |
| Grasslands | 92.38 | 79.66 |
| Green-colour roofs | 81.69 | 90.64 |
| Grey roofs | 70.38 | 69.02 |
| Red roofs | 93.43 | 94.97 |
| Tarred roads | 53.47 | 77.42 |
| Trees | 75.96 | 72.51 |

### 3.4. Comparison of 30 M Resolution with Sharpened 10 M Resolution LST Retrievals

Although the study area was small, variations in temperature were observed as some places were more than 15 °C cooler than others. Hotspots were noticed, especially on the southern half of the area where LSTs close to 49 °C were observed. The northern half was generally cooler, with the dominance of LSTs close to 41 °C. There was a general southeastward warming in the area. Comparison of Figure 4a,b shows that sharpening of LSTs to 10-m resolution by blending Landsat-derived LSTs with 10-m resolution Sentinel 2 did not compromise the spatial structure of LST and their ranges in the area. The 30-m resolution LST map was more pixelated than the 10-m resolution, indicating the latter's

improved quality. When compared to the 30-m resolution, 10-m LST were retrieved with high accuracy (RMSE = 0.5 °C). Correlation between LULC and NDVI was −0.516 and −0.999 before and after pan-sharpening, respectively.

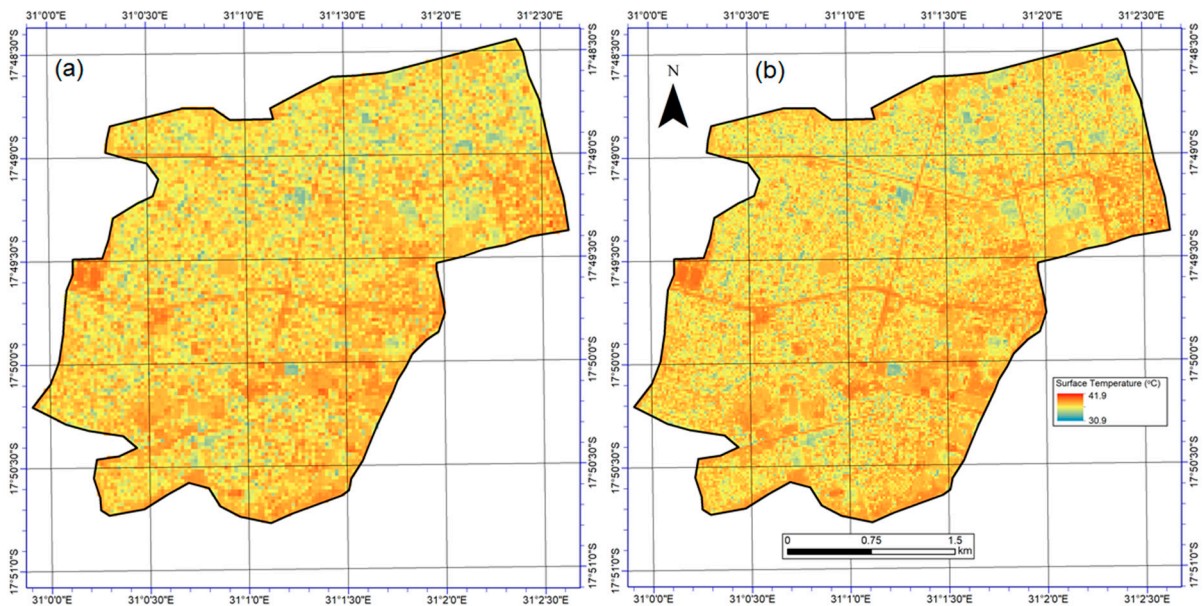

**Figure 4.** Spatial structure of (**a**) LST derived from Landsat thermal data at 30-m resolution (**b**) LST sharpened to 10-m resolution.

### 3.5. Variations of LST with LULC and Roof Colours

Although there were overlaps in temperature between different LULCs and roof colour categories, their mean LSTs were clearly distinct (Figure 5). The mean LST was lowest in the trees LULC category followed by blue roof. Highest LSTs were recorded in green-colour roofs and tarred roads areas. The grasslands LCZ showed greatest variability in LST, followed by green and red roofs. The order of roof colours from coolest to warmest based on average LST was blue (36.2 °C), black (35.8 °C), grey (36.9 °C), red (37.4 °C) and green (37.7 °C).

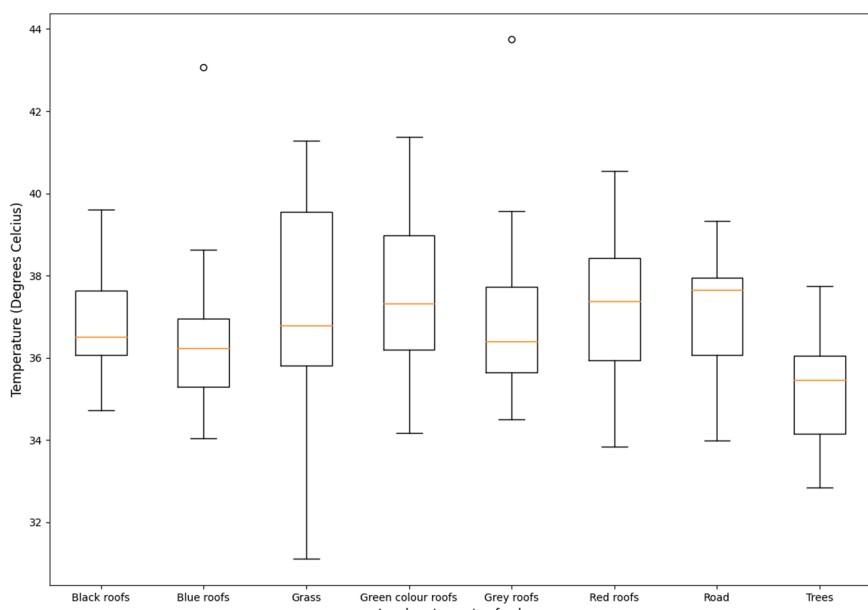

**Figure 5.** Observed variations of LST with LULC and roof colour classes in the study area.

## 4. Discussion

Separability of all classes was high, as indicated by a TDSI greater than 1.7. This was attributed to the strength of the spectral information at the 10-m resolution bands of Sentinel 2 to distinguish between different LULC and roof colours. As aforementioned, separability values close to 2 indicate that the classes are sufficiently separable using the remotely sensed image guided by GCPs [49]. The LULCs and roof colours in the study area were mapped with 75% and 0.73 accuracy and kappa, respectively. To facilitate effective separability of the heterogeneous study area, the study focused on a small spatial extent. Hence, the mapping accuracy was reasonable and comparable with other studies in urban environments such as Sithole and Odindi [62]. However, in this study, roads, (producer accuracy < 50%) were not effectively mapped. The low roads-mapping accuracy can be attributed to their narrow width, that they are largely below 10-m and along the road, and tree and tree shading, hence camouflage and/or mixed pixel with adjacent features.

Based on validation points, the roof colours were retrieved with reasonable accuracies. Producer accuracies (PA) ranged between 58 and 95%, while user accuracies (UA) were between 55 and 91%. The PA and UA values between 55 and 65% could be attributed to intra-class variabilities, which caused some similarities between different roof colours. For instance, some fading shades of black were near similar to dark shades of grey. Similarly, some shades of blue were closer to grey and black. Although not investigated, we speculate that roof ages and fading influence the similarities in roof colours. This is consistent with Alchapar and Correa [46] who noted that for a given roof colour, thermal properties can change due to age. Mapping accuracy could also be influenced by other effects such as roof shapes, reflectivity [63] and ventilation. For instance, Triano-Juárez et al. [64] observed variations in thermal properties for the same roof colour depending on reflectivity and presence of coating materials. On the other hand, Bojić et al. [65] observed differences between slanted and flat roofs. However, despite the above-named factors that could influence thermal variability based on roof colouring, our study shows that roof colours could be mapped with acceptable accuracy. We however suggest that for applications that require very high mapping accuracy (>90%), the Sentinel 2′s 10-m resolution data may be insufficient. In this regard, the use of Unmanned Aerial Vehicles derived high spatial resolution data offers great potential for fine-scale mapping.

Similar LST spatial structure was observed before and after sharpening, while accuracy of retrieved 10-m resolution LST relative to the original 30-m resolution was high (RMSE of about 0.5 °C). Similar to a recent study by Mushore et al. [28], pan-sharpening also improved correlation between LST and NDVI. In this study, the LST maps effectively showed thermal variations. Spatial comparison of the LULC and LST maps showed that vegetation covers such as large grasslands and trees as well as built-up areas with abundant vegetation (which characterize most of the study area) had comparatively low temperature, an indication that even vegetation within built-up areas has heat mitigation value [62]. Zhang et al. [66] also highlighted that vegetation patches and spatial structure combine in contributing to the reduction in surface temperature of the area they occupy. This explains the surface-temperature-reduction effect of vegetation even within built-up areas. Besides latent heat transfer, the shading effect of vegetation, especially trees, lowers surface temperatures in areas they cover. As such, Zhao et al. [48] noticed the cooling effect of shadows of surrounding trees on roof-top surface temperatures during daytime.

The grey and red roofs were warmer than the black roofs, but cooler than green-colour roofs, which were the warmest (Figure 4). Contrary to expectation, black roofs were not the warmest. This could be attributed to variations in thermal characteristics in relation to, among others, roof and colour shading. For example, due to age, black roofs colouring ranged between dark black and grey. Red and green also had higher thermal values. This finding is consistent with Farhan et al. [44], who found that red roofing had higher thermal values than white roofing. Our findings show that green-colour roofs were the warmest, with average LST values close to tarred roads. On the other hand, blue roofs were the coolest, a finding consistent with Libbra et al. [45], who note that roof colour influences

surface temperatures and hence could be used to mitigate heat islands. Although not investigated in this study, pigments on roof materials could have influenced their thermal behaviour. For example, it was reported that for the same roof colour, cool pigments have the potential to increase albedo by at least 20% [67]. This may have caused dark roofs to absorb less than or comparable heat to light-coloured roofs.

## 5. Conclusions

The 10-m resolution Sentinel 2 data mapped LULC and roofs by colour with reasonable accuracy. However, findings show that Sentinel 2′s 10-m spatial resolution is still limited by the mixed pixel problem. Other roof characteristics such as age, shape and coating need to be investigated for potential improvement in mapping accuracy. Sharpening of LSTs derived from Landsat to Sentinel's 10-m resolution improved the LST spatial structure. It also increased the correlation between LST and NDVI, implying an improved relationship with LULC. Different roof colour showed variations in mean LST, which highlighted the contribution of roof colours in mitigating or intensifying the heat island effect. Due to variations in shades attributed to changes in age, black roofs were not the warmest. Blue roofs were found to be the coolest while green-colour roofs were the warmest, followed by red roofs. Grey roofs had a moderate effect, with the cooling effect increasing with lightness of the grey colour. Overall, the study showed that colour, in combination with other roof properties, determines a building unit's thermal characteristics. However, the study observed that even after pan-sharpening, Sentinel 2′s 10-m spatial resolution was still coarse for urban roof mapping.

The study observed that even after pan-sharpening, Sentinel 2s 10-m spatial resolution was still coarse for urban roof mapping. This implies the need to test other higher spatial resolution datasets, for example those derived from UAVs and aircraft platforms. Future studies should also consider separating different shades of the same colour, especially in view of colour changes associated with roof aging. Additionally, the combined effects of various physical factors, which include roof coating, thickness, ventilation, and shape, should be included for in-depth analysis of the effect of roofs on the area's thermal environment. Among the factors to be included simultaneously is the presence and effect of any pigment that may affect albedo and heat absorption capacities, even for rooftops of the same colour. Given the inadequacy of freely available moderate-resolution Landsat 8 and Sentinel datasets in mapping thermal properties of rooftops, there is a need to test other higher spatial resolution datasets, for example those derived from UAVs and aircraft platforms.

**Author Contributions:** Conceptualization, T.M., O.M. and J.O.; methodology, T.M.; software, T.M.; validation, T.M., J.O. and O.M.; formal analysis, T.M.; investigation, T.M., O.M. and J.O.; resources, T.M., O.M. and J.O.; data curation, T.M., O.M. and J.O.; writing—original draft preparation, T.M.; writing—review and editing, T.M., O.M. and J.O.; visualization, T.M.; supervision, O.M. and J.O.; project administration, O.M. and J.O.; funding acquisition, T.M., O.M. and J.O. All authors have read and agreed to the published version of the manuscript.

**Funding:** The research of this article was supported by DAAD within the framework of the climapAfrica program of the Federal Ministry of Education and Research. The publisher is fully responsible for the content. The work and article processing charge was also funded by the National Research Foundation of South Africa (NRF) Research Chair in Land Use Planning and Management (Grant Number: 84157).

**Data Availability Statement:** Remotely sensed data used in this study can be freely downloaded from Earth Explorer website (www.earthexplorer.usgs.gov) courtesy of United States Geological Survey (USGS). Accessed on 10 January 2022.

**Acknowledgments:** We acknowledge the Climate Modeling Group of the climapAfrica fellowship for the support and the Discipline of Geography at the University of KwaZulu-Natal and the Department of Space Science and Applied Physics at the University of Zimbabwe for providing conducive working environment.

**Conflicts of Interest:** The authors declare no conflict of interest.

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
