# Peer review of "“Cool” Roofs as a Heat-Mitigation Measure in Urban Heat Islands: A Comparative Analysis Using Sentinel 2 and Landsat Data"

_remotesensing, doi:10.3390/rs14174247_

Round 1
Reviewer 1 Report
The study " Urban heat island and "cool" roofs: a comparison of roof colors as a heat mitigation measure using Sentinel 2 and Landsat data." examined used the freely available Sentinel 2 and Landsat 8 data to determine the study area's Land Use Land Covers (LULCs), roof colors and Land Surface Temperature (LST) at a 10m spatial resolution by using Support Vector Machines (SVM) classification algorithm
1. Currently, this manuscript contains 24% plagiarism, along with 7% from the below-mentioned work, which is so high from a single publication and unacceptable.
Terence et al. (2022) "Pansharpened Landsat 8 thermal-infrared data for improved Land Surface Temperature characterization in a heterogeneous urban landscape", Remote Sensing Applications: Society and Environment, https://doi.org/10.1016/j.rsase.2022.100728
2. As per the journal's requirement research article should be at least 16 pages, but this manuscript currently contains 16 pages, which may require an extension. The academic editor can have a look at this issue.
3. Line 26-26. I am confused about the roof colors and their Colling effects; the Authors mentioned "Black roofs" in the study area were cool while Green roofs were the warmest. In general, dark-colored roofing materials (e.g., Black) absorb the heat and cause a warming house than lighter colored material (e.g., green, white, etc.). Authors can further have a look at this work https://newscenter.lbl.gov/2004/08/27/cool-colors-cool-roofs/
4. Line 55-58. Consider adding a reference or citing correctly.
5. As per the limitations reported in lines 84-85. Please recheck the below-mentioned research and elaborate on the novelty of this work and why it should be published in remote sensing, where several advanced works already have been published in this domain
· Zhao, Q.; Myint, S.W.; Wentz, E.A.; Fan, C. Rooftop Surface Temperature Analysis in an Urban Residential Environment. Remote Sens. 2015, 7, 12135-12159. https://doi.org/10.3390/rs70912135
· https://www.sciencedirect.com/science/article/abs/pii/S0360132311002472
· Ejiagha, I.R.; Ahmed, M.R.; Hassan, Q.K.; Dewan, A.; Gupta, A.; Rangelova, E. Use of Remote Sensing in Comprehending the Influence of Urban Landscape's Composition and Configuration on Land Surface Temperature at Neighbourhood Scale. Remote Sens. 2020, 12, 2508. https://doi.org/10.3390/rs12152508
· Estimation of the relationship between vegetation patches and urban land surface temperature with remote sensing https://doi.org/10.1080/01431160802549252
6. Line 143. Please elaborate on the hot and dry seasons for a better understanding.
7. Section 2.2 dealt with remotely sensed datasets, but I could not find any preprocessing methodology for the datasets there. On the other hand, the authors didn't perform any preprocessing.
8. For section 2.5, follow the below-mentioned work for a clear description of LST retrieval
Mumtaz, F.; Tao, Y.; de Leeuw, G.; Zhao, L.; Fan, C.; Elnashar, A.; Bashir, B.; Wang, G.; Li, L.; Naeem, S.; Arshad, A.; Wang, D. Modeling Spatio-Temporal Land Transformation and Its Associated Impacts on land Surface Temperature (LST). Remote Sens. 2020, 12, 2987. https://doi.org/10.3390/rs12182987
9. Line 253. Please recheck with https://www.usgs.gov/media/images/landsat-8-band-designations. The spatial resolution of a thermal band of landsat8 is 100m, not 30m
10. Consider adding a methodology flowchart to understand mythology flow better.
11. Discussion of this work still has some room for modifications and can be modified.
12. It's better to add a sub-section 5.2. "Limitation and future aspects" to discuss the limitations of this work.
13. Conclusion can be modified.
Author Response
Kindly find file attached with response to reviewers

Reviewer 2 Report
The manuscript of Mushore et al. brings some interesting observation on the role of roof colors for the mitigation of urban heat island. The study is limited to Harare(Zimbabwe) and has potential to be publisher in Remote Sensing journal after taking to account the below comments.
Major comments:
The authors should explain how the different moment during the day when the Landsat and Sentinel images were taken could affect the final results. More than this, more images and also nighttime and daytime satellite images could build a more robust idea on the tackled topic. In the current form the results are interesting but not very sound.
L152-154: The meteorological conditions during selected days should be presented in details in order to understand if those days were characterized by (ab)normal temperature conditions. The results could be influenced by these precise weather conditions.
Specific comments:
The title could be reworded as: “Cool” roofs as a heat mitigation measure in urban heat island: a comparison using Sentinel 2 and Landsat data.
I recommend the transformation of the values presented in the results from Kelvin to Celsius.
Please be more precise when you talk about “green roofs”. It is not very clear in the manuscript if you refer here to roofs with vegetation or to green color painted roofs.
Minor comments:
L17: “was used derive the study area”. Please rephrase, this is not clear at all.
L87: focused on
L89: a dot after al is needed
L94: that the use
L121: Give the full name of acronym at its first use
L133: Figure 1 is repeated in the title caption
L391-393: The given example on snow is futile for Zimbabwe.
L405: Co-relation should be something different than co-relation?
Graphical aspects:
The resolution of the figures is (very) poor. As authors have chosen to work on a quite limited area it is expected that the spatial details will be very high, which is not the case in the manuscript. I suggest to redesign all the graphic materials.
Also, English related aspects should be refined. Often, the text is hard to read. Many phrases are too long and with an unnecessary degree of complexity.
Author Response

(The authors gave the same response as above.)

Round 2
Reviewer 1 Report
Authors' responses are satisfactory